# Performance of Textile Mask Materials in Varied Humidity: Filtration Efficiency, Breathability, and Quality Factor

**Joelle M. Segovia** [1], **Ching-Hsuan Huang** [2], **Maxwell Mamishev** [3], **Nanhsun Yuan** [1], **Jiayang He** [1] and **Igor Novosselov** [1,*]

1   Department of Mechanical Engineering, University of Washington, Seattle, WA 98195, USA
2   Department of Environmental and Occupational Health Sciences, School of Public Health, University of Washington, Seattle, WA 98195, USA
3   Nathan Hale High School, Seattle, WA 98125, USA
*   Correspondence: ivn@uw.edu

**Abstract:** During the COVID-19 pandemic, reusable masks became ubiquitous; these masks were made from various fabrics without guidance from the research community or regulating agencies. Though reusable masks reduce the waste stream associated with disposable masks and promote the use of masks by the population, their efficacy in preventing the transmission of infectious agents has not been evaluated sufficiently. Among the unknowns is the effect of relative humidity (RH) on fabrics' filtration efficiency (FE) and breathability. This study evaluates the FE and breathability of several readily accessible mask materials in an aerosol chamber. Sodium chloride aerosols were used as the challenge aerosol with aerodynamic particle diameter in the 0.5 to 2.5 μm range. To mimic the variability in RH in the environment and the exhaled-breath condition, the chamber was operated at RH of 30% to 70%. The face velocity was varied between 0.05 m/s and 0.19 m/s to simulate different breathing rates. The FE and pressure drop were used to determine the quality factor of the materials. Among the tested materials, the 3M P100 filter has the highest pressure drop of 140 Pa; the N95 mask and the 3M P100 have almost 100% FE for all sizes of particles and tested face velocities; the surgical mask has nearly 90% FE for all the particles and the lowest pressure drop among the certified materials, which ranks it the second to the N95 mask in the quality factor. Other material performance data are presented as a function of relative humidity and aerosol size. The quality factor for each material was compared against reference filtration media and surgical masks. Multiple layers of selected materials are also tested. While the additional layers improve FE, the pressure drop increases linearly. Additionally, the certified materials performed approximately three times better than the highest performing non-certified material.

**Keywords:** mask; aerosol; filtration efficiency; breathability; quality factor

## 1. Introduction

Respiratory infections are the most common illnesses and are one of the leading causes of mortality worldwide [1,2]. Infection transmission comes from the fomite route and exposure to infectious aerosols. Airborne transmission can cause large outbreaks even when individuals have minimal contact with fomites [3–5]. The airborne transmission mechanism involves infectious aerosols in particle diameter ($d_p$) ranging from 0.01 to 100 μm [6]. Typically, larger particles ($d_p > 5$ μm) have been classified as droplets, whereas those with $d_p < 5$ μm are classified as aerosols [7]. Aerosols remain suspended in the air for hours, long enough to be inhaled, and can contain multiple viral copies [8–10].

Aerosol inhalation can be reduced by using a face mask as the material provides filtration of inhaled and exhaled air, protecting the users and those around them. Filtration occurs via five different mechanisms: interception, inertial impaction, diffusion, electrostatic attraction, and gravitational settling [11,12]. The effectiveness of these filtration mechanisms

depends on the particle size and the type of filtration media. Gravitational settling is prevalent in removing droplets $d_p > 50$ μm. Inertial impaction is primarily seen on aerosols larger than 1 μm, whereas diffusion primarily affects the smallest particles [12]. As the flow rate increases, so does the effect of interception and inertial impaction, while the gravitational settling, diffusion, and electrostatic attraction become less impactful.

Face masks should provide the wearer with high filtration efficiency (FE), while remaining comfortable. Mask comfort can be attributed to the fit of the mask, a mask's ability to transfer heat and moisture away from the face, and its breathability [13]. Breathability is described by the permeability and pressure drop across the material [13,14]. Part of the certification testing for mask materials that public health organizations, such as the National Institute for Occupational Health (NIOSH), conduct is testing a materials pressure drop. Materials may not exceed the pressure drop values at the corresponding flow rate to ensure the breathability of the mask. Respirators can reach pressure differentials ranging from 210 to 350 Pa, whereas surgical masks should range between 40 Pa/cm$^2$ and 70 Pa/cm$^2$ [14,15]. At the start of the COVID-19 pandemic, high demand resulted in a scarcity of masks [16–19], leading to many using homemade masks made from various textiles. However, the protection from aerosolized transmission provided by these masks remains unclear, and the public has not received clear guidance from health officials.

Previous studies compared the performance of homemade masks and those certified by organizations such as NIOSH [20–44]. However, the accuracy of the results and the methodology used by some of these studies have come under scientific scrutiny [14,45–47]. Some studies did not provide sufficient details on the experimental methodology. For example, one study erroneously stated that N95 respirators did not provide 95% FE when tested with no leakage points [21,46,47]. Certified masks are subjected to a standardized testing procedure to ensure their ability to filter out 95% of aerosols. The NIOSH testing procedure uses a TSI Respirator Fit Tester to test twenty filters. With extensive testing procedures, masks that receive a certified N95 rating have a high-accuracy standard and excellent consistency between the samples. Real-time particle sizing instruments can significantly simplify and expedite the test procedures. An increasing number of studies use real-time particle sizers for calculating collection and transmission aerosols [48–51], for sensor calibration studies [52–56], and recently became a well-accepted method for testing the filtration efficiency, e.g., refs [57–63].

Environmental conditions and breathing rates affect local relative humidity (RH) in the fiber matrix. In a high RH environment, hygroscopic fibers absorb water and swell [64], affecting the material's porosity and pressure drop [65]. Few studies have examined the effects of humidity on the overall performance of textiles. The "quality factor" index can be used to parameterize this effect; it describes a mask's performance based on the ability to filter out particulates and the breathability [11].

This paper aims to complement previous studies describing the FE of common mask materials while adding new insight into the effects of RH on a material's filtration performance. In our study, materials were challenged with polydisperse sodium chloride (NaCl) aerosols in a size range of ~0.3 to 2.5 μm. The face velocity through the material was varied at 0.05 m/s, 0.10 m/s, and 0.19 m/s to simulate the variability in the breathing rate [15]. RH was varied in the 30% to 70% range. The FE and the pressure drop were measured to determine the quality factor. The certified materials by NIOSH outperform other textiles by a factor of three. Additionally, it is found that humidity has minimal effects on both the FE and the breathability of the materials in the tested RH range.

## 2. Materials and Methods

### 2.1. Materials

The materials tested in this study range from the masks certified by NIOSH, natural, synthetic, blended fabrics, and other non-traditional materials such as Thinsulate, heavyweight surgical wrap, and coffee filters. Certified masks include a 3M disposable N95 respirator filter, a 3M P100 filter, and a disposable 3-ply surgical mask (BYD care).

Fabrics tested in this study consisted of knitted cotton, muslin, and Kona cotton, along with silk and rayon. One natural-synthetic fabric blend was also tested, which was 82% rayon and 18% knitted cotton. Finally, more non-traditional materials were tested due to their commercial accessibility and their use as mask materials. These materials consisted of Thinsulate, heavyweight spunbond-meltblown-spunbond (SMS) polypropylene, and paper coffee filters. Optical microscopy images of these materials are shown in Figure 1, and additional details on each material are shown in Table S1 of the Supplemental Information.

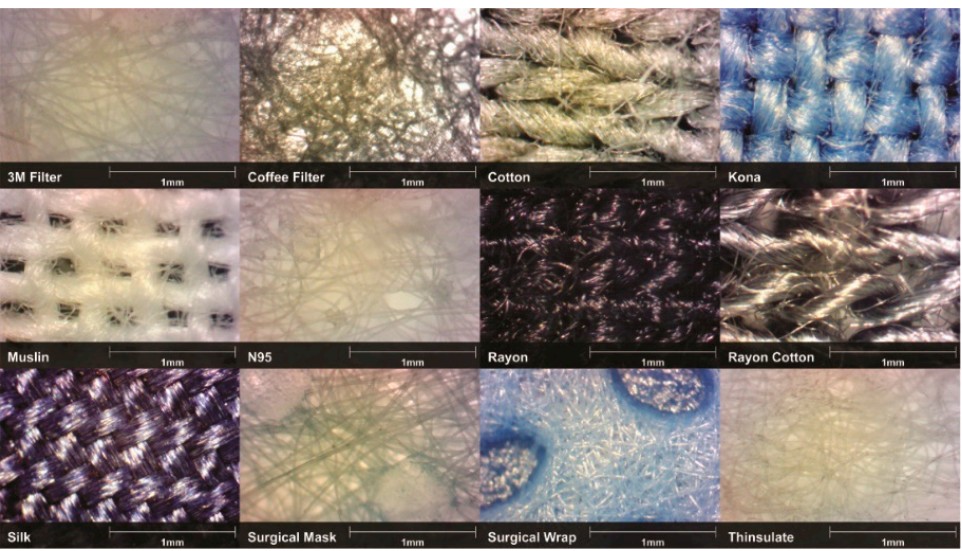

**Figure 1.** Microscopic images of the tested materials. Photos are taken at 2× magnification and are shown with a 1 mm scale.

The multiple material layers were evaluated, as double masking becomes a common practice for people to get more protection, affecting the FE, pressure drop and overall quality factor of the mask. The materials selected for testing with additional layers were the surgical mask, muslin, and coffee filter. The surgical mask was tested with two layers, whereas the muslin and coffee filter materials were tested with two and three layers.

### 2.2. Aerosol Chamber Setup

The experiments were conducted using a custom-built aerosol chamber (0.56 m × 0.52 m × 0.42 m) [66] with two 3D printed material holders attached. Two medical nebulizers (VixOne Small Volume Jet Nebulizer, Westmed, Tuscon, AZ, USA) were placed in the chamber. One of the nebulizers was used to generate polydisperse NaCl aerosols. The aerosolization was stopped when the particle concentration reached 900 to 1200 particles/cc, measured by the Aerodynamic Particle Sizer (APS 3321, TSI, Shoreview, MN, USA). The second nebulizer controlled the RH inside the chamber set to RH of 30%, 50%, or 70%. The two mixing fans inside provided homogeneous particle distribution in the chamber [2,66]. A 3D rendering of the aerosol chamber and the position of these components are shown in Figure 2(left).

Attached to the aerosol chamber are two sample holders: one holds the textile sample, and the other is empty, serving as a reference channel. Magnetic clips were used for alignment, and two binder clips were used to ensure an airtight seal, which was verified before each experiment. Particle-laden air from the aerosol chamber was aspirated through the material holder at a rate of 1 L per minute (LPM) from the APS. An additional make-up flow was provided by the building vacuum line at flow rates from 0.1 LPM to 3 LPM to reach the designed face velocity [50]. The make-up air was filtered by an in-line high-efficiency particulate air (HEPA) filter. A differential manometer (UEi EM201B) measured the pressure drop. The APS was connected to the holders through anti-static tubing to minimize deposition through electrostatic attraction.

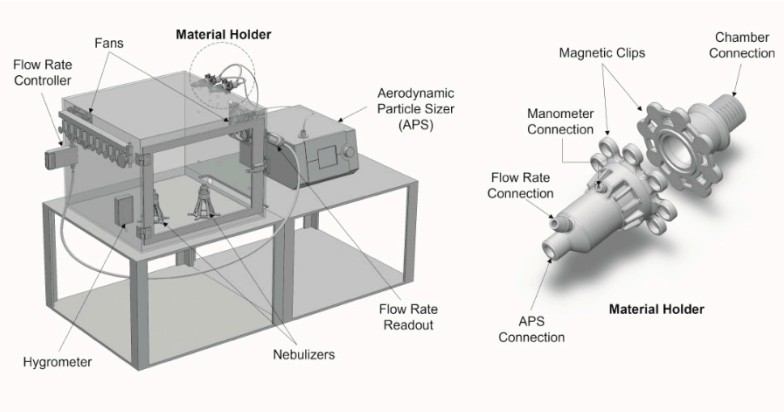

**Figure 2.** 3D render of the experimental setup. A custom-built aerosol chamber (**left**) contains two mixing fans, two nebulizers, and a hygrometer. Nebulizers generate polydisperse NaCl aerosols and control the humidity within the chamber. Attached to the chamber are two sample holders (**right**), one containing a material insert and one left empty to be used as a reference. The containers are connected to a mass flow rate controller and a manometer to measure the pressure drop. The flow passes through the material holders (one at a time) to the APS for data collection.

### 2.3. Pressure Drop and Filtration Efficiency Analysis

The breathability of the material was measured in terms of pressure drop. Pressure drop readings were taken over a range of face velocities between 0 m/s to 0.39 m/s and at each humidity level. Readings were taken using the differential manometer (UEI Test Instruments) connected to the material holder that measures the pressure difference across the sample material. Three pressure drop measurements for each face velocity were taken to determine a standard deviation.

To calculate FE, we took six measurements at each humidity and face velocities of 0.05 m/s, 0.10 m/s, and 0.19 m/s. Each reading was taken over ten seconds, alternating between filtered and reference streams. *FE* is calculated using Equation (1):

$$FE = \frac{n_{ref} - n_f}{n_{ref}} * 100\% \tag{1}$$

where the $n_{ref}$ and $n_f$ are the size-resolved aerosol concentrations measured by the APS for each size bin in the reference and in the filtered stream. The data collected from the APS is recorded in particle mass concentrations with an assumed density of 1.03 g/cm$^3$ for NaCl and in number density for the smaller particle bin ($d_p$ = 0.3–0.52 μm). Though the APS records the particle in the range $d_p$ = 0.3–20 μm, the data did not show a significant concentration of particles > 2.5 μm. To evaluate the FE as a function of particle size, we have binned the data into $d_p$ = 0.3–0.52 μm, $d_p$ < 0.97 μm (PM1), and $d_p$ < 2.46 μm (PM2.5). Any calculations showing a negative FE due to the reference and filtered measurements variance are reported as "zero." The three replicates are averaged for each condition providing the standard deviation shown as error bars in the plots in the result section.

The filter quality factor Q combines the material's FE and pressure drop describing its overall performance [11]. By combining the FE and the pressure drop, the desired functions of a mask, such as comfort and high filtration abilities, can be presented by a single value. The best filter is the one that has the highest FE with the lowest pressure drop. The *Q* is calculated using the following equation:

$$Q = \frac{ln\left(\frac{1}{1 - \frac{FE}{100}}\right)}{Pressure\ Drop} \tag{2}$$

## 3. Results and Discussion

### 3.1. Pressure Drop

Figure 3 shows pressure drop for three certified mask materials as a function of face velocity for three RH levels. The P100 3M filtration media had the highest pressure drop, and the surgical mask had the lowest. Several materials (muslin, knitted cotton, and rayon cotton blend) had lower pressure drops than a surgical mask. A complete list of materials' pressure drops can be found in the Supplemental Material Table S1. Pressure drop has a linear relationship to the face velocity through the material, as expected for the laminar airflow through the filter media. Not considering the dust load, the pressure drop ($\Delta P$, Pa) is proportional to the air velocity at the face of a filter ($U_0$, m/s). $\Delta P = \beta U_0$, where $\beta$ is the air resistance coefficient (Pa·s/m) [11].

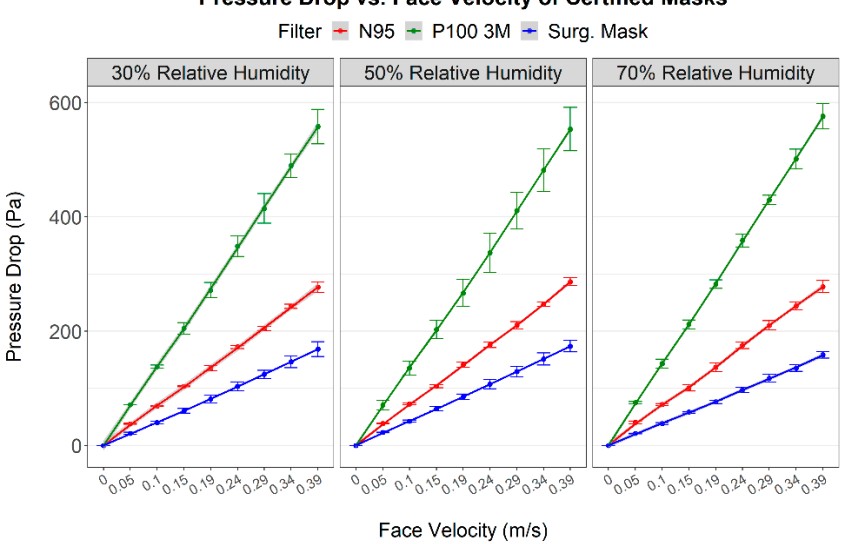

**Figure 3.** Pressure drop for single-layered certified materials as a function of face velocity. Facets represent the relative humidity during the experiment. Results are plotted as the average pressure drop from three runs, with the error bars representing the standard deviation.

Figure 4 shows the change in pressure drop of single-layer materials when the RH is varied. The effect of RH on single-layer materials' pressure drop is minimal or, in some cases, statistically insignificant. The minor pressure-drop increase on some hygroscopic materials such as silk, coffee filter, and cotton can be attributed to fiber swelling, a decrease in fabric porosity, and a possible increase in fabric thickness. A similar change in the resistance to convective flow at varying humidity levels was reported by Gibson, where the largest change was observed at higher RH above 0.8 [67]. Commercial surgical face masks typically have a three-layer structure. The middle layer is made of a melt-blown material that serves as the filter media, whereas the inner layer is for absorbing moisture, and the outer layer repels water. The moisture repelling by the outer layer might explain the small change in pressure drop and FE of the surgical mask at elevated RH levels. For multiple material layers, humidity has minimal to no effects on the breathability of the layered materials, as shown in Figure S2 of the Supplemental Information. As expected, the material's pressure drop increased with each additional textile layer.

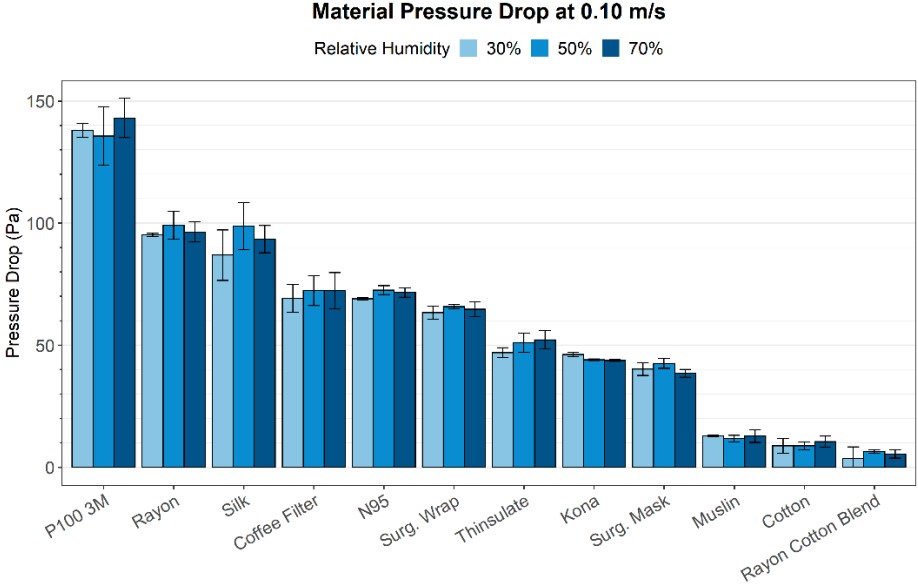

**Figure 4.** Pressure drop for single-layered materials as a function of RH. Bar colors represent the corresponding RH value, and results are plotted as the average pressure drop from three runs, with the error bars representing the standard deviation.

### 3.2. Filtration Efficiency

Figure 5 shows the filtration efficiency of materials at different face velocities for PM2.5, PM1, and particles in $d_p$ = 0.3–0.52 μm range. The certified masks had the highest efficiency of all particle sizes; P100 and N95 had 100% flirtation for all particle channels and tested face velocities. Surgical mask filtration was lower, especially for particles smaller than 0.5 μm; however, the surgical mask outperformed all other tested materials for homemade masks. The high FE of the certified materials for the ultrafine particles is due to electrostatically charged fibers densely woven by the melt-blown extrusion process. The mechanical filtration mechanisms (e.g., inertial deposition, interception, and diffusion) are combined with electrostatic deposition to filter both large and small particles. Commonly used textiles, such as cotton and cotton blends, had filtration efficiency below 10%. More dense Kona cotton approached 25% FE; however, its pressure drop was 3 to 10 times greater than other textiles and about the same as the surgical mask. Synthetic materials showed increased FE at lower face velocities for smaller particles. This trend indicates that the primary filtration mechanisms for PM1 are diffusion and electrostatic attraction, as synthetic fibers are likely to carry permanent electrostatic charges or be charged by triboelectrification, which improves the deposition of small particles [27,68,69]. However, certified multilayer material such as surgical masks has clear advantages due to the removal of moisture in the outer layer and the strong attraction of electrostatically charged fibers. Natural fiber textiles did not show this trend, and in the case of the paper coffee filter, higher velocity resulted in higher filtration for all particle sizes.

Figure 6 shows the FE for multiple-layered materials. As the aerosol sizes increase, the effects from interception and inertial impaction filtration mechanisms begin to appear, increasing the FE with increasing face velocity. The trend between FE and face velocity seen by the single-layer materials continued as additional layers were added. Additional layers provided a higher FE, as seen by the surgical mask, which required two layers to reach a consistent FE above 95%. These results represent the FE when a mask tightly fits the wearer with no leakage. Masks that are not properly fitted would not be able to provide the same level of FE, making the fit of masks a vital component to ensure maximum protection for the wearer [27,40,70].

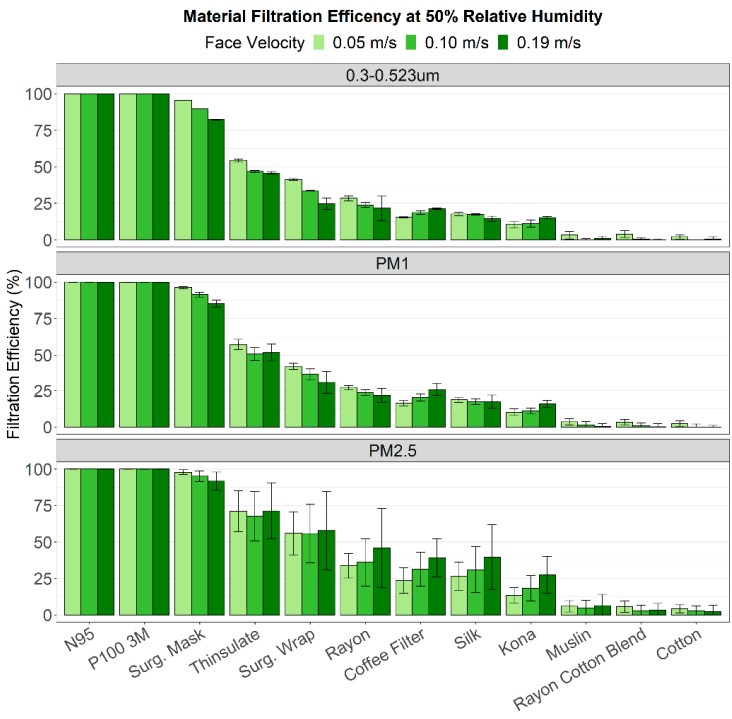

**Figure 5.** FE results for single-layered materials as a function of face velocity. Bar colors represent the corresponding face velocity values, with the results plotted at the average FE from three experimental runs with the error bars representing the standard deviation.

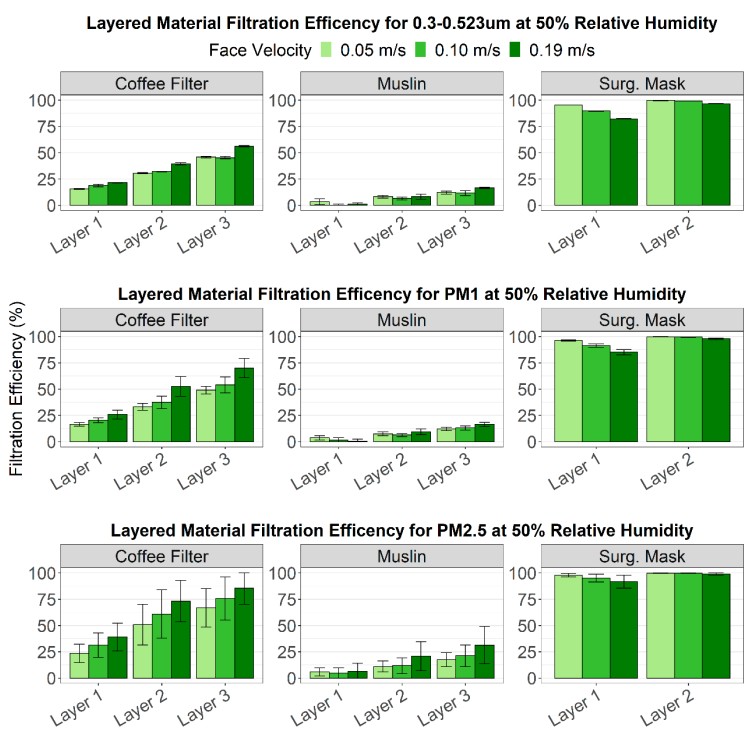

**Figure 6.** FE results for layered materials as a function of face velocity. Bar colors represent the corresponding face velocity values; the results are plotted at the average FE from three measurements, with the error bars representing the standard deviation.

*3.3. Quality Factor*

The overall performance of a material is quantified by using the quality factor to relate the FE and breathability a material provides. Figure 7 shows the quality factor of tested materials at three face velocities for PM2.5. The quality factors for other particle sizes can be found in the Supplemental Information. The quality factor was calculated using three measurements' average FE and pressure drop readings. In terms of the quality factor, the top three performing materials in this study were those commonly worn as PPE: the N95 mask, the surgical mask, and the P100 3M filter. Even though the N95 and P100 3M filters are considered more difficult to breathe through, their high FE makes them considerably better materials when compared to non-traditional mask materials. These certified materials were found to have, at a minimum, a quality factor three times higher than the top non-traditional mask material. The critical differences are the utilization of electrostatically charged fibers and the hydrophobic layers that are not considered in the homemade masks. Natural fabrics are all found to have the lowest quality factor ranging from 0.01 to 0.02. The effect of multiple material layers on the quality factor was also analyzed and included in the supplemental material. The quality factor tended to decrease as additional layers were added. While materials, such as the surgical mask, provided higher FE when double layers were used, the additional pressure from the second layer caused the overall mask performance to decline, as shown in Figure S5 of the Supplemental Information.

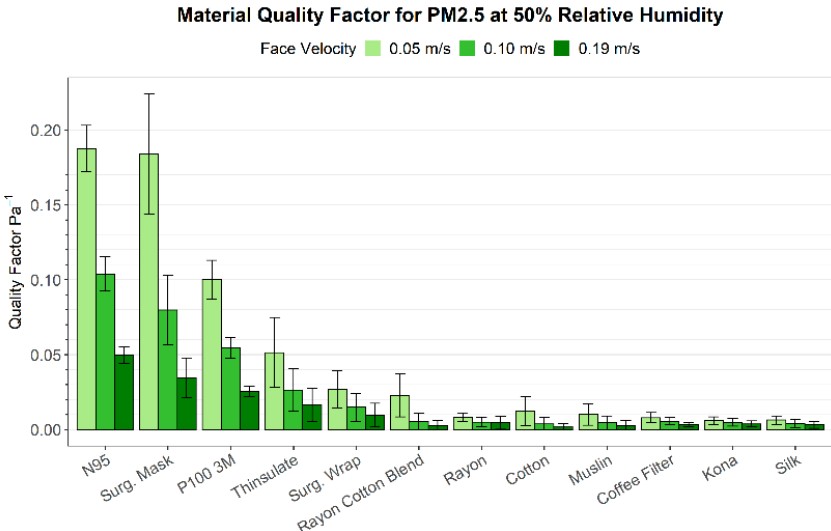

**Figure 7.** Quality factors for single-layered materials when face velocity is varied. Error bars in the figure represent the standard deviation across these three quality factor values, while the bar colors represent the corresponding face velocity values.

## 4. Conclusions

This study found that mask performance is dependent on the face velocity through the material rather than the relative humidity. Though varying humidity can inhibit particle growth and hygroscopic behavior in certain materials, the effects from humidity on mask FE and pressure drop are insignificant. While the impact of improper mask fittings is not analyzed in this study, ensuring the mask fits with no leakage will supply the wearer with filtered air within the filtration efficiencies shown in this study. In terms of overall performance, masks certified by health organizations such as NIOSH perform at least three times better than other non-certified materials. Non-certified materials provide the wearer with some protection and comfort; however, they do not provide as much protection as certified masks. While adding additional layers of material can increase a mask's FE, they can become increasingly difficult to breathe through, outweighing the increase in aerosol protection. These results suggest that single-layered surgical masks should be recommended, given their high FE, comfortability, and accessibility. Clips or additional

accessories should be used to create a proper fit, reducing the risk of leakage. Higher-grade surgical masks should be used by those in areas where they are subjected to a high amount of air contaminants and users should refrain from long-term wear.

**Supplementary Materials:** The following supporting information can be downloaded at: https://www.mdpi.com/article/10.3390/app12189360/s1, Figure S1: Photos of some of the tested materials. Listing the materials starting from the top left and moving across are: Kona, Muslin, Cotton, Rayon, Rayon Cotton blend, and Surg. Wrap; Figure S2: Pressure drop results for layered materials as a function of RH. Bar colors represent the corresponding RH value and results are plotted as the average pressure drop from three runs, with the error bars representing the standard deviation; Figure S3: Filtration efficiency results for single layered materials as a function of relative humidity. Bar colors represent the corresponding RH values with the results plotted at the average filtration efficiency from three experimental runs with the error bars representing the standard deviation; Figure S4: Filtration efficiency results for layered materials as a function of relative humidity. Bar colors represent the corresponding RH values with the results plotted at the average filtration efficiency from three experimental runs with the error bars representing the standard deviation.; Figure S5: Quality factor results for layered materials when face velocity is varied. The quality factor results use the average filtration efficiency and pressure drop readings taken from three experiments to calculate three individual quality factor values. Error bars in the figure represent the standard deviation across these three quality factor values while the bar colors represent the corresponding face velocity values; Table S1: Material list, description, and average filtration efficiency, pressure drop, and quality factor values. Materials are listed from highest quality factor to lowest. Average and standard deviation values are calculated across three experiments.

**Author Contributions:** Conceptualization, J.H. and I.N.; methodology, J.H. and I.N.; software, J.M.S. and N.Y.; formal analysis, J.M.S. and C.-H.H.; data curation, N.Y. and J.M.S.; writing—original draft preparation, J.M.S. and M.M.; writing—review and editing, J.M.S. and J.H.; visualization, J.M.S.; supervision, I.N. and J.H.; project administration, J.H. and I.N.; funding acquisition, I.N. All authors have read and agreed to the published version of the manuscript.

**Funding:** This research received no external funding.

**Conflicts of Interest:** The authors declare no conflict of interest.

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
