# Peer review of "Performance of Textile Mask Materials in Varied Humidity: Filtration Efficiency, Breathability, and Quality Factor"

_applsci, doi:10.3390/app12189360_

Round 1

Reviewer 1 Report

This paper is interesting and innovative for academic area of built environment. While the subject of the analysis is very meaningful and the content of the article is very abundant, the manuscript needs to be revised before accepted for publication. My detailed comments are as follows:

1. In the Abstract and Conclusion, the authors are suggested to add some quantitative findings in this study.

2. I would strongly suggest to develop the Results section further. For example, the authors should try to explain why does the humidity have effects on the pressure drop and filtration efficiency? In the current version, only findings are reported without detailed analysis.

3. The authors tried many different kinds of material in this study. The authors are suggested to explain the reasons for the different performance between different materials.

Author Response

This paper is interesting and innovative for academic area of built environment. While the subject of the analysis is very meaningful and the content of the article is very abundant, the manuscript needs to be revised before accepted for publication. My detailed comments are as follows:

Point 1: In the Abstract and Conclusion, the authors are suggested to add some quantitative findings in this study.

Response 1: Thank you for your suggestion. We have added some quantitative findings and updated the abstract as follow:

During the COVID-19 pandemic, reusable masks became ubiquitous; these masks were made from various fabrics without guidance from the research community or regulating agencies. Though reusable masks reduce the waste stream associated with disposable masks and promote the use of masks by the population, their efficacy in preventing the transmission of infectious agents has not been evaluated sufficiently. Among the unknowns is the effect of relative humidity (RH) on fabrics' filtration efficiency (FE) and breathability. This study evaluates the FE and breathability of several readily accessible mask materials in an aerosol chamber. Sodium chloride aerosols were used as the challenge aerosol with aerodynamic particle diameter in the 0.5 to 2.5 µm range. To mimic variability in RH in the environment and the exhaled breath condition, the chamber was operated at RH of 30% to 70%. The face velocity was varied between 0.05 m/s and 0.19 m/s to simulate different breathing rates. The FE and pressure drop were used to determine the quality factor of the materials. Among the tested materials, the 3M P100 filter has the highest pressure drop of 140 Pa; the N95 mask and the 3M P100 have almost 100% FE for all sizes of particles and tested face velocities; the surgical mask has nearly 90% FE for all the particles and the lowest pressure drop among the certified materials, which ranks it the second to the N95 mask in the quality factor. Other material performance data are presented as a function of relative humidity and aerosol size. The quality factor for each material was compared against reference filtration media and surgical masks. Multiple layers of selected materials are also tested. While the additional layers improve FE, the pressure drop increases linearly. Additionally, the certified materials performed approximately three times better than the highest performing non-certified material.

Point 2: I would strongly suggest to develop the Results section further. For example, the authors should try to explain why does the humidity have effects on the pressure drop and filtration efficiency? In the current version, only findings are reported without detailed analysis.

Response 2: Thank you for the comment. We revised section 3.1, 3.2, and 3.3 to add more explanation of the RH’s effects on the FE and the pressure drop.

3.1. Pressure Drop

Figure 3 shows pressure drop for three certified mask materials as a function of face velocity for three RH levels. The P100 3M filtration media had the highest pressure drop, and the surgical mask had the lowest. Several materials (muslin, knitted cotton, and rayon cotton blend) had lower pressure drops than a surgical mask. A complete list of materials' pressure drops can be found in the supplemental material Table 1. Pressure drop has a linear relationship to the face velocity through the material, as expected for the laminar airflow through the filter media. Not considering the dust load, the pressure drop (ΔP, Pa) is proportional to the air velocity at the face of a filter (U0, m/s). ΔP = βU0, where β is the air resistance coefficient (Pa·s/m) [11].

Figure 4 shows the change in pressure drop of single-layer materials when the RH is varied. The effect of RH on single-layer materials' pressure drop is minimal or, in some cases, statistically insignificant. The minor pressure drop increase on some hygroscopic materials such as silk, coffee filter, and cotton can be attributed to fiber swelling, a decrease in fabric porosity, and a possible increase in fabric thickness. A similar change in the resistance to convective flow at varying humidity levels was reported by Gibson, where the largest change was observed at higher RH above 0.8 [70]. Commercial surgical face masks typically have a three-layer structure. The middle layer is made of a melt-blown material that serves as the filter media, whereas the inner layer is for absorbing moisture, and the outer layer repels water. The moisture repelling by the outer layer might explain the small change in pressure drop and FE of the surgical mask at elevated RH levels. For multiple material layers, humidity has minimal to no effects on the breathability of the layered materials, as shown in Figure S2 of the supplemental information. As expected, the material's pressure drop increased with each additional textile layer.

3.2. Filtration Efficiency

Figure 5 shows the filtration efficiency of materials at different face velocities for PM2.5, PM1, and 0.3 to 0.52 µm particles. The certified masks had the highest efficiency of all particle sizes; P100 and N95 had 100% flirtation for all particle channels and tested face velocities. Surgical mask filtration was lower, especially for particles smaller than 0.5 µm; however, the surgical mask outperformed all other tested materials for homemade masks. The high FE of the certified materials for the ultrafine particles is due to electrostatically charged fibers densely woven by the melt-blown extrusion process. The mechanical filtration mechanisms (e.g., inertial deposition, interception, and diffusion) are combined with electrostatic deposition to filter both large and small particles. Commonly used textiles, such as cotton and cotton blends, had filtration efficiency below 10%. More dense Kona cotton approached 25% FE; however, its pressure drop was 3 to 10 times greater than other textiles and about the same as the surgical mask. Synthetic materials showed increased FE at lower face velocities for smaller particles. This trend indicates that the primary filtration mechanisms for PM1 are diffusion and electrostatic attraction, as synthetic fibers are likely to carry permanent electrostatic charges or be charged by triboelectrification which improves the deposition of small particles [28,71,72]. However, certified multilayer material such as surgical masks has clear advantages due to the removal of moisture in the outer layer and the strong attraction of electrostatically charged fibers. Natural fiber textiles did not show this trend, and in the case of the paper coffee filter, higher velocity resulted in higher filtration for all particle sizes.

3.3. Quality Factor

The overall performance of a material is quantified by using the quality factor to relate the FE and breathability a material provides. Figure 7 shows the quality factor of tested materials at three face velocities for PM2.5. Quality factors for other particle sizes can be found in the supplemental information. The quality factor was calculated using three measurements' average FE and pressure drop readings. In terms of the quality factor, the top three performing materials in this study were those commonly worn as PPE: the N95 mask, the surgical mask, and the P100 3M filter. Even though the N95 and P100 3M filters are considered more difficult to breathe through, their high FE makes them considerably better materials when compared to non-traditional mask materials. These certified materials were found to have, at a minimum, a quality factor three times higher than the top non-traditional mask material. The critical differences are the utilization of electrostatically charged fibers and the hydrophobic layers that are not considered in the homemade masks. Natural fabrics are all found to have the lowest quality factor ranging from 0.01 to 0.02. The effect of multiple material layers on the quality factor was also analyzed and included in the supplemental material. The quality factor tended to decrease as additional layers were added. While materials, such as the surgical mask, provided higher FE when double layers were used, the additional pressure from the second layer caused the overall mask performance to decline, as shown in Figure S5 of the supplemental information.

Point 3: The authors tried many different kinds of material in this study. The authors are suggested to explain the reasons for the different performances between different materials.

Response 3: Please see our response to comment 2 for the reasons for the different performances.

Reviewer 2 Report

This paper is very well written. It experimentally investigated the filtering efficiency and breathability (the two factors are combined to form the quality factor). The setup is simple, and the results seem sound enough. I don’t have any substantial objection for the publication of this paper.

However, the analysis and discussion in the paper is not “advanced” enough. Only simple comparisons are provided. The relevant discussion about the filtration mechanism is not sufficient. The RH condition turned out not to be an important factor, and this conclusion compromised the significance of this research. Personal experience shows that masks can become quite “wet” during using. How about the FE under such conditions?    

Line 153, remove indention

Line 175, right parenthesis is missing

Figs 5 and 6, resolutions are too low, error bars are broken.

Author Response

This paper is very well written. It experimentally investigated the filtering efficiency and breathability (the two factors are combined to form the quality factor). The setup is simple, and the results seem sound enough. I don't have any substantial objection for the publication of this paper.

However, the analysis and discussion in the paper is not "advanced" enough. Only simple comparisons are provided. The relevant discussion about the filtration mechanism is not sufficient. The RH condition turned out not to be an important factor, and this conclusion compromised the significance of this research. Personal experience shows that masks can become quite "wet" during using. How about the FE under such conditions?    

Response: Thank you for your comments. We added more analysis and discussions in the result section. Please see the attachment for the updated manuscript. Due to our setup's limitation, we could only get approximately 70% RH. No significant difference in FE was observed. However, we noticed that a similar change in the resistance to convective flow at varying humidity levels was reported by Gibson, where the largest change was observed at higher RH above 0.8. We speculate that RH >80% might have more impact on the FE because of more mechanical filtration caused by fiber swelling.

Point 1: Line 153, remove indention

Response 1: The indentation was removed.

Point 2: Line 175, right parenthesis is missing

Response 2: The right parenthesis was added.

Point 3: Figs 5 and 6, resolutions are too low, error bars are broken.

Response 3: Fig 5 and Fig 6 were replaced with high-resolution figures.

Round 2

Reviewer 1 Report

This manuscript has been well revised. It can be received.